# Identifying the sources of uncertainty in Object Classification

## Abstract

In image-based object classification, the visual appearance of objects determines which class they are assigned to. External variables that are independent of the object, such as the perspective or the lighting conditions, can modify the object's appearance resulting in ambiguous images that lead to misclassifications. Previous work has proposed methods for estimating the uncertainty of predictions and measure their confidence. However, such methods do not indicate which variables are the potential sources that cause uncertainty. In this paper, we propose a method for image-based object classification that uses disentangled representations to indicate which are the external variables that contribute the most to the uncertainty of the predictions. This information can be used to identify the external variables that should be modified to decrease the uncertainty and improve the classification.

## 1 Introduction

An object from the real world can be represented in terms of the data gathered from it through an observation/sensing process. These observations contain information about the properties of the object that allows their recognition, identification, and discrimination. In particular, one can obtain images from objects which represent its visual characteristics through photographs or rendering of images from 3D models.

Image-based object classification is the task of assigning a category to images obtained from an object based on their visual appearance. The visual appearance of objects in an image is determined by the properties of the object itself (intrinsic variables) and the transformations that occur in the real world (extrinsic variables) (Kulkarni et al., 2015).

Probabilistic classifiers based on neural networks can provide a measure for the confidence of a model for a given prediction in terms of a probability distribution over the possible categories an image can be classified into. However, they do not indicate what variable contributes to the uncertainty. In some cases the extrinsic variables can affect the visual appearance of objects in images in such way that the predictions are highly uncertain. A measure of the uncertainty in terms of these extrinsic features can improve interpretability of the output of a classifier.

Disentangled representation learning is the task of crating low-dimensional representations of data that capture the underlying variability of the data and in particular this variability can be explained in terms of the variables involved in data generation. These representations can provide interpretable data representations that can be used for different tasks such as domain adaptation (Higgins et al., 2017),continuous learning (Achille et al., 2018), noise removal (Lopez et al., 2018), and visual reasoning (van Steenkiste et al., 2019).

In this paper we propose a method for the identification of the sources of uncertainty in image-based object classification with respect to the extrinsic features that affect the visual appearance of objects in images by using disentangled data representations. Given an image of an object, our model identifies which extrinsic feature contributes the most to the classification output and provides information on how to modify such feature to reduce the uncertainty in the predictions.

## 2 RELATED WORK

Achieving explainable results in predictive models is an important task, especially for critical applications in which the decisions can affect human life such as in health, security, law and defence Barredo Arrieta et al. (2020). Even though deep neural networks provide successful results for image classification, their predictions can't be directly interpreted due to their complexity (Zhang & Zhu, 2018). In order to solve this different approaches have been proposed to provide visual interpretability to the results such as identification of the image regions that contribute to classification (Selvaraju et al., 2016) .

The uncertainty of predictions provides an extra level of interpretability to the predictions of a model by indicating the level of confidence in a prediction Kendall & Gal (2017). There are different methods to introduce uncertainty measures in classifiers which include bayesian neural networks, ensembles, etc.

Obtaining disentangled representations, that capture distinct sources of variation independently, is an important step towards interpretable machine learning systems Kim & Mnih (2018). Despite the lack of agreement on the definition, one description states that a disentangled representation should separate the distinct, informative factors of variations in the data Bengio et al. (2012).

Within deep generative models, disentanglement is achieved by using neural networks that approximate a conditional distribution on the data and a set of unobserved latent variables. Particularly variational autoencoders (VAEs) Kingma & Welling (2014) are heavily favored due to their ability to model a joint distribution while maintaining scalability and training stability Higgins et al. (2016). Therefore most of the methods are based on augmentations on original VAE framework Higgins et al. (2016); Kim & Mnih (2018); Kulkarni et al. (2015); Mathieu et al. (2018) .

In image-based object classification the variables that explain the visual characteristics of objects in data can be divided into those which represent the inherent properties of objects and those which represent its transformations. This explanation has been explored in Kulkarni et al. (2015) by describing the object's properties as the intrinsic variables and the properties that describe the object transformations as the extrinsic variables.

Other work refers to similar sets of variables and their disentanglement under different names but representing similar concepts. Hamaguchi et al. (2019) disentangles the variables corresponding to ambient variables with respect to object identity information on images. (Gabbay & Hoshen, 2020) proposes the learning of disentangled representations that express the intra-class variability in terms of the class and content. (Detlefsen & Hauberg, 2019) proposes the disentanglement of the appearance and the perspective. Separate the identity of cars from their pose (Yang et al., 2015).

## 3 SETTING

Consider a dataset of images that have been generated from the observations of an object and which should be classified into a certain category. We will assume that this category depends only on the properties of the object itself and not on its surroundings.

We use a neural network as a probabilistic classifier to assign each of the images to a category. Usually the output of a neural network can't be directly interpreted in terms of the characteristics of the object have affected the confidence of the prediction. Disentanglement serves as a method to produce interpretable low-dimensional data representations that separate the variables that describe the properties of the objects and their surrounding.

The main idea is that one can train a probabilistic classifier on disentangled low dimensional representations and identify which variables contribute to the uncertainty of the classification.

### 3.1 PROBABILISTIC CLASSIFIERS ON IMAGES

A probabilistic classifier is a model which outputs a conditional probability density $P_{Y|x}$ over the set of $K \in \mathbb{N}$ possible categories $Y = \{1, 2, \ldots, K\}$ conditioned on an input image $x \in X$. The value $P_{Y|x}(y)$ can be interpreted as the degree of confidence the model assigns for an image $x \in X$

to be classified into category $y \in Y$. We will be only considering throughout this work probabilistic classifiers that use deep neural networks to obtain the predictions.

One can train a probabilistic classifier using a dataset of labeled images. Given a labeled datapoint $x \in X$ with true label $y^* \in Y$, the neural network's weights are optimized to reduce the cross entropy between the network's output distribution $P_{Y|x}$ and the true label $y*$. The cross entropy loss corresponds to,

$$\mathcal{L}(P_{Y|x}, y*) = -\sum_{y \in Y} \delta_{y,y*} \log P_{Y|x}(y), \tag{1}$$

with $\delta_{y,y*}$ the Kroenecker delta.

One can measure the degree of uncertainty of the output by calculating the entropy of the output distribution $P_{Y|x}$ for a given image. Higher entropy corresponds to a higher uncertainty. The entropy is calculated as,

$$H(P_{Y|x}) = \sum_{y \in Y} P_{Y|x}(y) \log P_{Y|x}(y). \tag{2}$$

Even though it is possible to provide a degree of uncertainty to the estimates of a probabilistic classifier trained on images, these estimates do not indicate which true generative variables that describe the image have contributed to the uncertainty. In this paper we assume that those sources of uncertainty are due to the extrinsic variables that participate in the data generation process.

### 3.2 Data Generation: Intrinsic and extrinsic variables

We will assume that the underlying generative process for our data can be modelled by the joint probability distribution $P_{X \times V}$ over the data space $X$ and a set of true generative variables $V$. The true variables condition the data generative process and represent the properties of the real world. In our case, we will consider that these variables can be divided into two sets $V = V^{(I)} \times V^{(E)}$ that represent the intrinsic and extrinsic variables respectively.

The intrinsic variables are those which represent the properties of the object itself (e.g. its color, shape, size, material), while the extrinsic variables represent external factors (e.g. the light conditions or the relative position and orientation of the object with respect to the observer that generates the image). Intrinsic variables are invariant to the transformations described by the extrinsic variables.

The generative process consists of the independent sampling of an intrinsic variable $v^{(I)} \sim P_{V^{(I)}}$ and an extrinsic variable $v^{(E)} \sim P_{V^{(E)}}$. Those generative variables together determine a conditional distribution over the data space from which a data point $x \sim P_{X|(v^{(I)},v^{(E)})}$ is sampled.

We assume that the intrinsic variables are independent of the extrinsic variables. During data generation both variables are combined to produce the visual appearance of the objects following the formula for the joint probability density

$$P_{X \times V}(x, (v^{(I)}, v^{(E)})) = P_{X|(v^{(I)},v^{(E)})}(x) P_{V_I}(v^{(I)}) P_{V_E}(v^{(E)}). \tag{3}$$

We assume that the set of intrinsic and extrinsic variables can be separated into a finite product of $M_I, M_E \in \mathbb{N}$ variables such that $V^{(I)} = V_1^{(I)} \times \cdots \times V_{M_I}^{(I)}$ and $V^{(E)} = V_1^{(E)} \times \cdots \times V_{M_E}^{(E)}$. The total number of true generative variables is then $M = M_I + M_E$.

### 3.3 Disentanglement for Interpretability

Learning disentangled representations is the task of creating interpretable low-dimensional data representations that separate the information about the variables that are involved in the generation of the data (Locatello et al., 2018). There is no common agreement on the definition of a disentangled representation. However, two properties have been proposed for disentangled representations that will be useful for our goal of producing interpretable measures of uncertainty in classification predictions.

- *Modularity*: No more than a single dimension of the data representation should encode information about a true generative variable.

- *Compactness*: Each of the true generative variables is encoded by a single dimension of the data representation.

If a data representation is both modular and compact, we can identify for each true generative variable its corresponding data representation's dimension that has all information about it. Consider $Z = Z_1 \times Z_2 \times \cdots \times Z_D$ as a $D$-dimensional data representation space. One can measure the compactness of a learned representation by measuring the mutual information between a data representation dimension in $Z_i$ and a true variable $V_m$ as

$$\mathcal{I}(P_{Z_i}, P_{V_m}) = \mathrm{KL}(P_{Z_i \times V_m} || P_{Z_i} \otimes P_{V_m})$$

A disentangled representation is modular and compact if for each generative variable $V_m$ there is a unique latent dimension $Z_i$ such that $\mathcal{I}(P_{Z_i}, P_{V_m}) \neq 0$ and $\mathcal{I}(P_{Z_j}, P_{V_m}) = 0$ for all $j \neq i$. If a disentangled data representation is obtained which fulfills both modularity and compactness then we can separate the latent dimensions in such a way that there is a decomposition of the latent space $Z = Z^{(I)} \times Z^{(E)}$ into an intrinsic and an extrinsic set of latent variable dimensions. This would mean that a probabilistic classifier trained on the latent variables $P_{Y|Z}(y) = P_{Y|Z^{(I)}}(y)$ would only depend on the intrinsic variables.

However, it has been proved in (Locatello et al., 2018; Mathieu et al., 2018) that without supervision disentanglement cannot be achieved. Therefore one might not be able to achieve perfect modularity or compactness. This means that some information about the true intrinsic variables might be contained in the extrinsic latent dimensions such that there is a dependency of the uncertainty in those variables as seen in Section 4.2.

## 3.4 Probabilistic Disentanglement

Probabilistic methods for disentanglement propose a generative distribution that approximates the true generative process in terms of a set of unobserved latent variables that serve as the data representation space $Z$. For simplicity we will assume that $Z$ corresponds to a $D$-dimensional vector space, e.g. $Z = \mathbb{R}^d$. The generative probability density function $P_{X \times Z}$ over the data space $X$ and latent space $Z$ can be expressed as

$$P_{X \times Z}(x, z) = P_{X|z}(x) P_Z(z). \tag{4}$$

In order to provide a good approximation to the true generative model, the data distribution determined by the true model and the one proposed in terms of the latent variables should match. This is expressed via the marginalization of the generative probability densities over the set of true generative variables and latent variables respectively, i.e.

$$P_X(x) = \int_V P_{X|v}(x) P_V(v) dv = \int_Z P_{X|z}(x) P_Z(z) dz. \tag{5}$$

One problem is that the integral above is in most cases intractable. There are different approaches to solve this problem. In particular, variational inference offers a method for approximating to the target true probability density (Blei et al., 2018) by means of an approximation to the true posterior distribution.

The variational autoencoder (Kingma & Welling, 2014) implements variational inference using an encoding neural network that calculates the parameters of the posterior approximate $Q_{Z|x}$ and the distribution $P_{X|z}$ also called the decoder distribution by using neural networks. The neural network weights are optimized by maximizing the lower bound of the latent variable model distribution over the data $P_X$.

Given a data point $x \in X$ and the approximate posterior $Q_{Z|x}$ it is possible to define a function over the data space $z : X \to Z$ such that for data point $x \in X$ its representation in the latent space is given by

$$z(x) = \mathbb{E}_{z \sim Q_{Z|x}}[z]. \tag{6}$$

### 3.5 Disentanglement for Probabilistic Classifiers

A disentangled representation can be used to train a probabilistic classifier and provide interpretability to the results in terms of each of its latent dimensions. Since the information about the true generative variables is encoded in the separate latent variables one can estimate the uncertainty of such classifier and measure the change in the uncertainty by small perturbations in the latent dimension.

We can train a probabilistic classifier with respect to the latent variables i.e. $P_{Y|\boldsymbol{z}(x)}$ and estimate the uncertainty of a prediction as $H(P_{Y|\boldsymbol{z}(x)})$. If this probabilistic classifier is trained with a neural network, we can obtain the gradient of the entropy with respect to the latent variables $\nabla_Z H(P_{Y|\boldsymbol{z}(x)})$ via backpropagation. The gradient indicates the direction in the latent space that leads to a lower entropy in the prediction, i.e. to a latent variable with lower uncertainty.

For a given image $x \in X$, it is possible to find the latent variable that would decrease the uncertainty by using the function $z^* : X \to Z$ defined by

$$z^*(x) = \boldsymbol{z}(x) - \alpha \nabla_Z H(P_{Y|\boldsymbol{z}(x)}). \tag{7}$$

where $\alpha$ corresponds to the size of the step taken in the direction of the negative gradient. Alternatively, it is possible to estimate the latent dimension that provides the highest change in the uncertainty of the prediction. We can evaluate the new latent variable obtained by taking a step along only the $d$-th latent dimension as

$$z'_d(x) = \boldsymbol{z}(x) - \alpha \frac{\partial H(P_{Y|\boldsymbol{z}(x)})}{\partial z_d} \hat{e}_d, \tag{8}$$

with $\hat{e}_d$ the $d$-th canonical basis vector. Then we can evaluate Equation 8 for each of the dimensions that corresponds to the extrinsic variables to find which is the one that provides the highest decrease in entropy

$$z^*(x) = \underset{z'_d \in Z^{(E)}}{\arg\max} \, H(P_{Y|\boldsymbol{z}(x)}) - H(P_{Y|z'_d(x)}) \tag{9}$$

The generative variable associated with the latent dimension that produces the highest decrease in entropy can be considered as the one that contributes the most to the entropy.

## 4 Experiments & Results

In order to test the ideas presented in the previous section we used a synthetic dataset where we could have control over a set of extrinsic and intrinsic variables. In this case we decided to use a dataset of images generated from 3D models where we could generate large amounts of data by varying different extrinsic variables.

**Modelnet40 Aligned Dataset**   In order to investigate whether the disentangled representations of data can be useful to identify the sources of uncertainty we have used a synthetic dataset where we have control over most of the variables involved in the data generation. In particular we generate a dataset of rendered images obtained from 3D models within the Modelnet40 aligned dataset which contains 3D models from 40 different classes. Each 3D model is aligned to a certain predefined orientation which is unique to each class. Images were generated from only the car class by rendering of the cars for different configurations of the camera and object properties: relative azimuth and elevation of the camera with respect to the car, the light intensity and location within the scene, see Figure 4. For the training set 40 car models were used and 8 were used for validation and 8 for test. We have manually labeled each of the cars in each of the datasets into four categories for classification: SUV, sport, sedan and hatchback.

For each 3D car model We generated a $6^5$ images by changing four extrinsic variables (azimuth, elevation, light intensity, location of light) and one intrinsic variable (color) over 6 possible values.

1. *Light Intensity:* Indicates the intensity of the light source used in Watts per square meter $W/m^2$. Values are from $\{0.5, 1.0, 1.5, 2.0, 3.0\}$

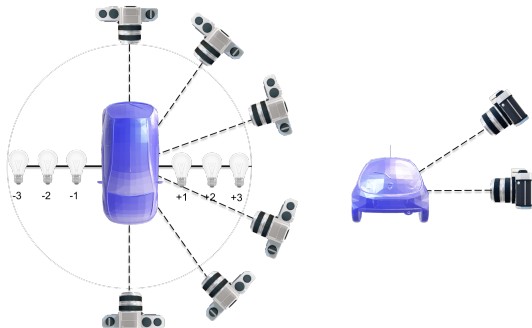

Figure 1: Diagram showing the changes in the extrinsic generative variables (light intensity, light location, elevation, azimuth) used to generate the dataset of images rendered from Modelnet40.

2. *Light Location:* Indicates the position of the light source along one axis, its values are $\{-3, -2, -1, 1, 2, 3\}$

3. *Elevation:* Determines the angular elevation of the camera with respect to the ground, the values used are $\{18.4°, 22.6°, 26.5°, 30.25°, 33.7°, 36.9°\}$.

4. *Azimuth:* Indicates the relative angle of rotation of the camera with respect to the front of the car when the camera is rotating with respect to an axis perpendicular to the floor, the values taken are $\{0°, 36°, 72°, 108°, 144°, 180°\}$.

5. *Color:* This variable corresponds to an intrinsic feature of the car. Different hues were chosen for each car: green, cyan, blue, magenta, red and yellow.

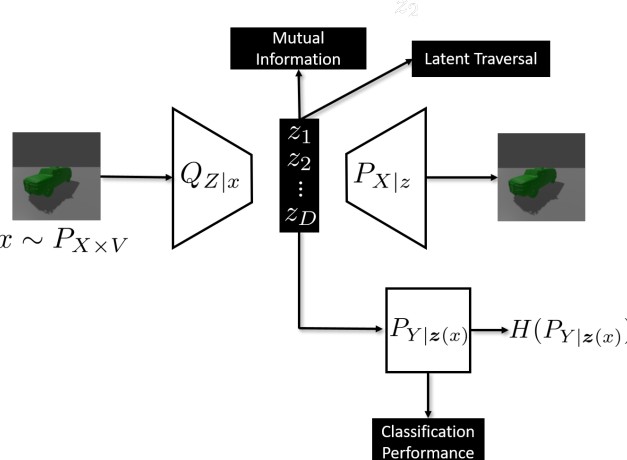

Figure 2: Diagram representing the encoding of car images into latent representations by using the DC-IGN encoder. The latent variables disentanglement is measured quantitatively via mutual information and qualitatively by doing latent traversals. The output classifier's uncertainty is calculated $H(P_{Y|\boldsymbol{z}(x)})$ based on its gradient it is possible to propose a new image that reduces the uncertainty by selecting the appropriate generative variables.

## 4.1 DISENTANGLED REPRESENTATIONS

We train a Deep Convolutional Inverse Graphics Network (DC-IGN) (Kulkarni et al., 2015) since it provides a method for obtaining disentangled representations by training a variational autoencoder Kingma & Welling (2014) with batches of images where only only one generative variable changes across the images. DC-IGN takes advantage of the batches to enforce the separation of the information of each generative variable into separate latent dimensions.

The encoder network has 3 convolutional blocks in which a convolutional layer with kernel size of 5 and $2 \times 2$ max pooling with stride 2 are used. Differently from the original implementation, we set the size of the latent space as 128. The decoder network consists of 3 convolutional blocks with a $2 \times 2$ upsampling layer each followed by convolutional layer with kernel size 7. The training is performed for a total of 100 epochs.

The disentangled representations were characterized by measuring the discretized mutual information as in Locatello et al. (2018). The results are shown in Table 4.2, latent traversals are presented in Figure 3 . Since the latent variables are encouraged to only contain information about the generative variable that changes in each batch we can directly associate a latent dimension to each generative variable as it can be seen in the results. The latent variable associated to the target generative variable does contain the largest value of mutual information.

One important note about our approach. we proposed the use of DC-IGN a disentanglement learning algorithm which is tailored for the separation of extrinsic features. However, it is possible to use any other disentanglement approach as long as there is a small subset of data where each extrinsic variable is available. In such case it is possible to measure the mutual information between each generative variable and latent dimension for the small dataset.

One can assign the latent dimension with the highest mutual information to each of the true generative variables such that the gradient with respect to a latent dimension indicates a change in the corresponding generative variable.

## 4.2 Probabilistic Classifier: Disentangled Representations

We train a probabilistic classifier with the latent variables obtained from the encoder DC-IGN by using the corresponding car type labels. We train a simple neural network with 2 dense layers with 150 and 50 neurons each respectively and an output dense layer with one neuron per category and a softmax activation function. The model is trained with early stopping using as hyperparameter a minimum loss decrease of $0.001$ and patience of 3 epochs. The classifier achieved $89\%$ accuracy on both the training and the test set. We estimated the uncertainty of the estimations using the softmax output of the intrinsic classifier. The average value of the metrics accross all car types is precision $0.91$, recall $0.89$, and F1-Score $0.89$.

We reasoned the validity of the predictive uncertainty by evaluating with the Expected Calibration Error (ECE) metric as described in Guo et al. (2017). The metric measures the expected error between the estimation confidences and the true class probabilities and indicates the discrepancies between the accuracy and the uncertainty measures in our case this gives a value of $0.015$ (lower is better calibration). Comparatively, as presented in Guo et al. (2017), ResNet 50 trained on Stanford Cars Krause et al. (2013) data set has $0.0430$ ECE without any additional calibration method. Even with additional calibration enchancements the lowest ECE of Resnet 50 set is $0.0174$.

Given that the intrinsic classifier is calibrated, the uncertainty of the classifier is estimated with the entropy of the softmax output. In Section 3.4 we mentioned that we expect the true extrinsic generative variables affect the uncertainty of a clasifier. To justify this claim, we measured the entropy of the images predicted for the classifier and compared their distribution across the values of each extrinsic generative variable.

If the extrinsic generative variables do not affect the uncertainty predictions then across different values of the same generative factor the distribution of entropies should be the same across different values of the extrinsic variable. We used the Mann-Whitney U (MWU) test to test whether the distribution of entropy between two different values of a generative variable are different. The null hypothesis for this test states that there is not a statistically significant difference between two distributions. In Appendix A we show the log entropy distributions for each extrinsic variable with violin plots together with the p-values from the pair-wise comparison of MWU test.

Only for the tests between angle $36°$ and $72°$ of the azimuth, $22.6°$ and $26.5°$ of elevation and $2, -2$ for the light location the null hypothesis cannot be rejected with a p-value larger than $0.05$. In other words except for these particular combinations of extrinsic variables we conclude that the distribution of entropy values are statistically different from each other for the generative variables. This means that the uncertainty of the predictions is indeed dependent on the true extrinsic variables for the trained classifier.

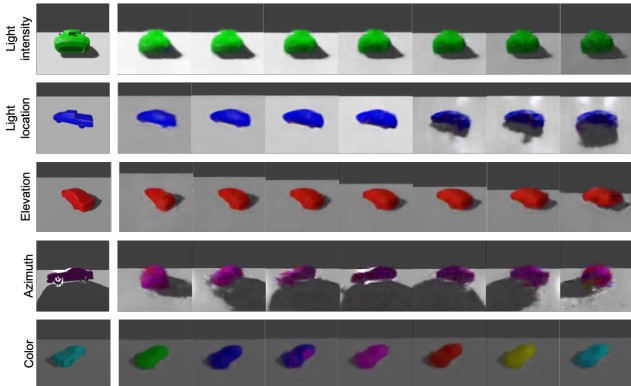

Figure 3: Examples of latent traversals across the latent dimensions correspond to each true generative variable.

Table 1: Mutual information values obtained between the corresponding latent dimensions of $Z$ associated to the corresponding true generative variables of $V$.

| $Z$ \ $V$ | Light Intensity | Light Location | Elevation | Azimuth | Color |
|---|---|---|---|---|---|
| Light Intensity | **1.230** | 0.104 | 0.010 | 0.019 | 0.008 |
| Light Location | 0.077 | **0.764** | 0.011 | 0.072 | 0.013 |
| Elevation | 0.071 | 0.022 | **1.562** | 0.024 | 0.007 |
| Azimuth | 0.098 | 0.067 | 0.011 | **0.938** | 0.010 |
| Color | 0.039 | 0.020 | 0.013 | 0.024 | **1.631** |

## 5 CONCLUSION

In this work we proposed the use of disentangled data representations to provide interpretability to the results of classifiers based on the set of extrinsic variables that affect the estimation of the intrinsic properties of objects.

We showed that for a probabilistic classifier trained on the disentangled latent variables, the extrinsic variables affect the uncertainty of the predictions. Moreover we provide a method to modify the latent variables in order to identify the extrinsic variable that produces the strongest uncertainty in the classifier's predictions.

Future work will consider the exploration of methods to react upon the sources of uncertainty in image-based object classification by proposing actions to an agent that gethers data in order to modify the extrinsic variables that produce the uncertainty.

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

## A   UNCERTAINTY DEPENDENCE ON EXTRINSIC VARIABLES

In this appendix, the plots representing the distribution of uncertainty values of the probabilistiic classifier trained on the disentangled latent variables together with the Mann-Whitney U test p-values are presented. The violin plots show the distribution of the uncertainty with respect to the true extrinsic variables and the p-value tables indicate whether a pair of values for an extrinsic variable produce statistically significant different uncertainty. If $p > 0.05$ the uncertainty for two values of a generative variable are similar.

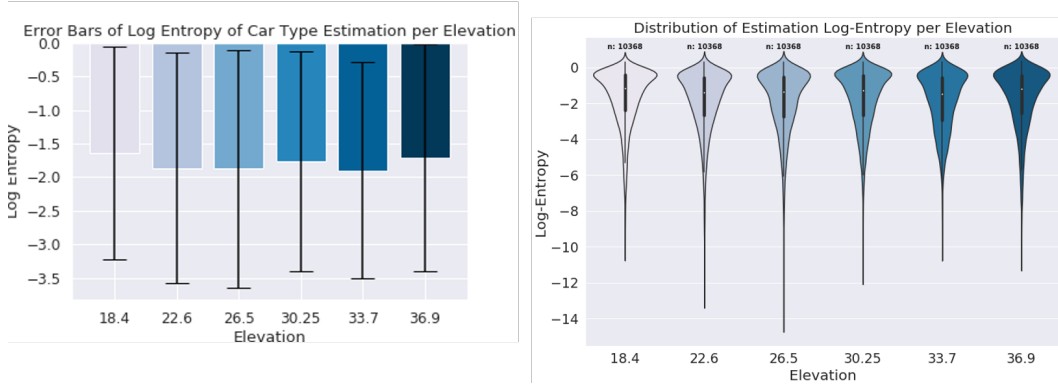

Figure 4: Error-bar and violin plots for the elevation extrinsic feature

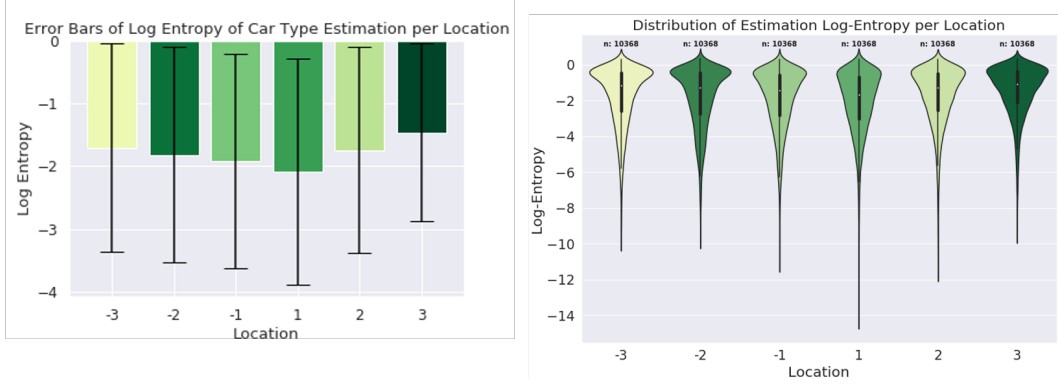

Figure 5: Error-bar and violin plots for the location extrinsic feature

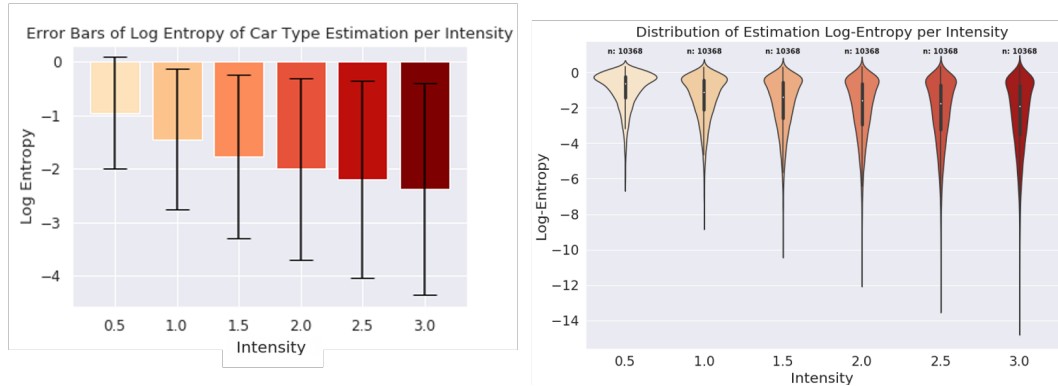

Figure 6: Error-bar and violin plots for the intensity extrinsic feature

| Azimuth | 0° | 36° | 72° | 108° | 144° | 180° |
|---|---|---|---|---|---|---|
| 0° | - | 0.000 | 0.000 | 0.000 | 0.000 | 0.000 |
| 36° | 0.000 | - | 0.190 | 0.000 | 0.000 | 0.001 |
| 72° | 0.000 | 0.190 | - | 0.000 | 0.000 | 0.003 |
| 108° | 0.000 | 0.000 | 0.000 | - | 0.000 | 0.000 |
| 144° | 0.000 | 0.000 | 0.000 | 0.000 | - | 0.000 |
| 180° | 0.000 | 0.001 | 0.003 | 0.000 | 0.000 | - |

Table 2: p-values of MWU test of azimuth latent variable

Table 3: p-values of MWU test for the location of elevation generative variable across different values

| Elevation | 18.4 | 22.6 | 26.5 | 30.25 | 33.7 | 36.9 |
|---|---|---|---|---|---|---|
| 18.4 | - | 0.000 | 0.000 | 0.000 | 0.000 | 0.045 |
| 22.6 | 0.000 | - | 0.198 | 0.000 | 0.000 | 0.000 |
| 26.5 | 0.000 | 0.198 | - | 0.000 | 0.000 | 0.000 |
| 30.25 | 0.000 | 0.000 | 0.000 | - | 0.000 | 0.000 |
| 33.7 | 0.000 | 0.000 | 0.000 | 0.000 | - | 0.000 |
| 36.9 | 0.045 | 0.000 | 0.000 | 0.000 | 0.000 | - |

Table 4: p-values of MWU test for the location of lightning true generative variable across different values

| Location | -3 | -2 | -1 | 1 | 2 | 3 |
|---|---|---|---|---|---|---|
| -3 | - | 0.000 | 0.000 | 0.000 | 0.000 | 0.000 |
| -2 | 0.000 | - | 0.000 | 0.000 | 0.142 | 0.000 |
| -1 | 0.000 | 0.000 | - | 0.000 | 0.000 | 0.000 |
| 1 | 0.000 | 0.000 | 0.000 | - | 0.000 | 0.000 |
| 2 | 0.000 | 0.142 | 0.000 | 0.000 | - | 0.000 |
| 3 | 0.000 | 0.000 | 0.000 | 0.000 | 0.000 | - |

Table 5: p-values of MWU test for the intensity of lighting true generative variable across different values

| Intensity | 0.5 | 1.0 | 1.5 | 2.0 | 2.5 | 3.0 |
|---|---|---|---|---|---|---|
| 0.5 | - | 0.000 | 0.000 | 0.000 | 0.000 | 0.000 |
| 1.0 | 0.000 | - | 0.000 | 0.000 | 0.000 | 0.000 |
| 1.5 | 0.000 | 0.000 | - | 0.000 | 0.000 | 0.000 |
| 2.0 | 0.000 | 0.000 | 0.000 | - | 0.000 | 0.000 |
| 2.5 | 0.000 | 0.000 | 0.000 | 0.000 | - | 0.000 |
| 3.0 | 0.000 | 0.000 | 0.000 | 0.000 | 0.000 | - |

