# OpenReview forum: "Identifying the Sources of Uncertainty in Object Classification"
_ICLR.cc/2021/Conference — Reject_

### Official Review · AnonReviewer3 · 2020-10-22

**Rating:** 3
**Confidence:** 4

**Review:**

Summary:

The paper presents an approach that for every object identifies the factors that have a high impact on the models' uncertainty. The approach consists of i) disentangled representations ii) a classifier on the top of the trained representations iii) technique that select dimensions of representation of an objects' (factors), that impact the uncertainty of a model the most. The i) and ii) are done in a known way; The disentanglement is done via Deep Convolutional Inverse Graphics Network (2015), and a classifier is trained with a standard maximum likelihood approach.
The iii) suggests selecting a dimension that influences uncertainty the most, based on the gradient of uncertainty criteria (entropy of p(y|z), where z is embedding). In other words, the suggestion is to select the component of representation that "provides the highest decrease in entropy".
The experiments are done in a controllable setting, where the generative process is known and can be controlled. The model demonstrated the ability to learn disentangled representations, in the case when control of generative prosses is available. Specifically, it is required to generate/have objects were "only one generative variable changes across the images". However, the connection to uncertainty is weak.

Review:

I have very mixed feelings about the work. On one hand, the problem is interesting (perhaps novel) and deserves the attention of the community. On the other hand, the paper is half-baked, there are conceptual flaws, the evaluation protocol is weak, there is an incorrect interpretation of experiment results;

My comment on novelty is the following: I would say that algorithmic novelty, is the weak point, however, bringing the attention of the community to important problems is not less important than new algorithms (and even is more important IMO). Taking into account that "novelty" is extremely subjective, I would say that there is nothing wrong with the novelty side.

Writing: the paper is clearly written.

Concerns:

1) The identified factors do not proofed to be meaningful and useful. Pure identifying factors of uncertainty or reduction uncertainty of a model do not give us the full picture. The factors may be flawed in many ways: factors may be too dependent on random errors of the model but not to be semantically meaningful. Reduction uncertainty of a model is not always useful, as most models nowadays are often wrongly overconfident, more confidence is not necessarily good! It has not been proven that identified factors are connected to uncertainty.

2) There is a noticeable connection between the proposed method and adversarial attacks. We can interpret a method as a way to "trick" a model to be more ceratin on adversarial modifications of embeddings. In this case, the results would not be useful.

3) There are two flaws of experiments in section 4.2: i) the authors compare Expected Calibration Error between different datasets, these figures are not comparable, it is the same as compare accuracy between MNIST and ImageNet  ii) ECE is a biased estimate of true calibration with a different bias for each model, so it is not a valid metric to compare even models trained on the same data (see Vaicenavicius2019). Yes, ECE is the standard in the field, but it is the wrong standard that prevents us from meaningful scientific progress, so we should stop using it.

Overall: The direction is interesting, however, the evaluation protocol needs to be more thoughtful. At the moment, it is impossible to verify the real performance of the method.

Suggestions:

1) Evaluate the identifying algorithm on downstream problems. For example, can we use these factors to collect additional data that will improve predictive performance better than uniformly sampled data (aka active learning)? There should be many more interesting settings.

2) Evaluate the model on controllable uncertainty estimation setting: identify setting where we understand on which examples we expect the model to be uncertain (due to not enough data in a dataset in a certain region, etc.), validate selected factors.

3) To use the squared kernel calibration error (SKCE) proposed in [Widmann2019] along with de facto standard, but biased ECE. The SKCE is an unbiased estimate of calibration. There might be some pitfalls of this metric that I'm not aware of, but the paper looks solid and convincing. Also, please put attention to Figure 83 in the arХiv version.

Editing:
- Citations: It is better to use "authors (year)" style when a citation is a part of a sentence---"(Gabbay & Hoshen, 2020) proposes" to "Gabbay & Hoshen, (2020) propose", and otherwise "(authors, year)" when a citation is not a part of a sentence "on original VAE framework Higgins et al. (2016);" to on original VAE framework (Higgins et al., 2016; .....).
- "only only" typo in 4.1

[Vaicenavicius2019] Juozas Vaicenavicius, David Widmann, Carl Andersson, Fredrik Lindsten, Jacob Roll, and Thomas B Schon. Evaluating model calibration in classification. AISTATS, 2019.

[Widmann2019] Widmann D, Lindsten F, Zachariah D. Calibration tests in multi-class classification: A unifying framework. In Advances in Neural Information Processing Systems 2019 (pp. 12257-12267). https://arxiv.org/pdf/1910.11385.pdf

---

### Official Review · AnonReviewer1 · 2020-10-27
**Addressing an important problem but lacks sufficient novelty and experimental validation**

**Rating:** 3
**Confidence:** 5

**Review:**

This paper proposes to identify sources of uncertainty by disentangling representations in latent spaces for object classification tasks. Experiments on a synthetic dataset demonstrate that the proposed method can disentangle different extrinsic variables' contribution to the prediction uncertainty.  In addition, the authors propose to modify the latent variables to decrease uncertainty in the predictions.

### Clarity
#### Pros
- This paper provides a clear discussion of the main contributions, assumptions, experimental settings.

#### Cons
- The experiment section needs more analysis. For example, how to explain the results in Table 1, which variable contributes most to the uncertainty and which contributes least?
- The paper contains typos and grammatical errors and need to be proofread carefully.
- Several questions needed to be addressed.

Questions:
1. How to choose the latent representations? In real applications, there are usually more factors.
2. It is not clear how the model in Figure 2 is trained. e.g. what is the loss function, do you need supervision on the latent variable.

### Originality
#### Pros
- The paper proposes a  method to identify sources of uncertainty by disentangling representations in latent spaces in object classification tasks.

#### Cons
The contribution is unclear as the disentangling method is based on the DC-IGN method.  Moreover, it is unclear how to relate the imaging factors to the latent variables as multiple imaging factors can impact the same latent variable.

### Significance
#### Pros
- It addresses an important problem; the problem of decomposition of the sources of uncertainty in model prediction is important and has not been sufficiently addressed in existing works.

#### Cons
- Entropy of the softmax is not a reliable criterion to capture uncertainty and, more importantly, other factors besides the imaging factors such as the input data density can also contribute to the uncertainty of the prediction.
- The experiment is performed on synthetic data, which leads to limited evaluation conditions. As there is a distribution difference between synthetic data and real data, it would be more significant if the proposed method can be applied to real images.
- The latent variables in the study are limited, only including light intensity, pose, color, etc. In real applications, there are usually many more factors such as background, occlusion, noise, resolution, etc. It would be better if the authors can study more factors or explain why specific factors are chosen.

---

### Official Review · AnonReviewer4 · 2020-10-28
**The method suggested does not apply to real world problems, and the findings presented are rather limited**

**Rating:** 3
**Confidence:** 3

**Review:**


Major comments:
-	The methods suggested does not apply to real-world classifiers:
o	It required training with strong labels only available in simulation
o	Classification is done based on a low-dimensional intermediate representation of the ‘entangled space’, and its accuracy is probably inferior to what can be obtained with general CNNs (no comparison was made)
o	I do not see how this can be applied to real-world competitive classifiers
-	The findings are rather limited
o	 The method suggested for finding which extrinsic variable is able to reduce the prediction entropy is a) not well founded, and b) was not tested experimentally
-	There are presentation clarity problems, with non-sentences, broken references, etc..

More detailed comments:

-	“Separate the identity of cars from their pose (Yang et al., 2015).” – this is not a sentence.
-	The probability estimates provided by neural networks are uncalibrated (not reflecting the real error probability)
               o	This is not mentioned in the introduction, but is discussed in page 7
-	Page 4 “maximizing the lower bound of the latent variable model distribution” – what is a lower bound over a distribution? Lower bounds are defined for scalars, not distributions. This sentence is not clear.
-	Page 5: before equation7 the text reads “it is possible to find the latent variable that would decrease the uncertainty…”. However, the equation does not define a latent variable, not does the text in the 1-2 sentences following it. The equation defines a vector in R^M. It is not stated how it defines a latent variable
-	Page 5: The process described in equations 8,9 is essentially making a gradient step (of the entropy) in each of the M_E possible directions and then measures which step reduced the Entropy. This is a numerical re-estimation of the gradient. Why is it done? At least for small \alpha, the direction most minimizing the entropy can be determined from the gradient directly (the coordinate in which the (negative of ) gradient is highest)
-	Page 7: The results are shown in Table 4.2, = 4.2 is a broken reference
o	Also: it is not clear how mutual information is calculated. It requires discrete variables, and the quantization details may be important.
-	The classifier trained has a bottleneck at the  ‘entangled representation’ layer, and is hence probably is far from being optimal in terms of its obtained error
-	The method suggested for estimating which extrinsic variables most affect the uncertainty was not applied experimentally (no results were reported)

---

### Decision · Program_Chairs · 2021-01-07
**Final Decision**

**Decision:**

Reject

**Comment:**

The manuscript presents an approach for identifying sources of uncertainty in object classification tasks by disentangling representations in latent spaces.

Three reviewers agreed that the manuscript is not ready for publication.
Some of the concerns are conceptual flaws, weak evaluation protocol, and an incorrect interpretation of experiment results.

There is no author response.